# Multivariate Analysis of Biochemical Properties Reveals Diversity among Yardlong Beans of Different Origins

**DOI:** 10.3390/antiox13040463

**Published:** 2024-04-14

**Authors:** Yu-Mi Choi, Hyemyeong Yoon, Myoung-Jae Shin, Sukyeung Lee, Jungyoon Yi, Young-ah Jeon, Xiaohan Wang, Kebede Taye Desta

**Affiliations:** 1National Agrobiodiversity Center, National Institute of Agricultural Sciences, Rural Development Administration, Jeonju 54874, Republic of Korea; cym0421@korea.kr (Y.-M.C.);; 2International Technology Cooperation Center, Technology Cooperation Bureau, Rural Development Administration, Jeonju 54875, Republic of Korea

**Keywords:** antioxidant activity, ascorbic acid, fatty acids, legumes, metabolomics, nutrients, *Vigna unguiculata*

## Abstract

This study analyzed the nutrient levels, secondary metabolite contents, and antioxidant activities of 35 yardlong bean accessions from China, Korea, Myanmar, and Thailand, along with their key agronomic traits. Significant variations were found in all the parameters analyzed (*p* < 0.05). The crude fiber (CFC), dietary fiber (DFC), total protein, and total fat contents varied from 4.10 to 6.51%, 16.71 to 23.49%, 22.45 to 28.11%, and 0.59 to 2.00%, respectively. HPLC analysis showed more than a 10-fold difference in vitamin C level (0.23 to 3.04 mg/g), whereas GC-FID analysis revealed the dominance of palmitic acid and linoleic acid. All accessions had high levels of total unsaturated fatty acids, which could help in preventing cardiovascular disease. Furthermore, total phenolic, tannin, and saponin contents ranged between 3.78 and 9.13 mg GAE/g, 31.20 and 778.34 mg CE/g, and 25.79 and 82.55 mg DE/g, respectively. Antioxidant activities like DPPH^•^ scavenging, ABTS^•+^ scavenging, and reducing power (RP) ranged between 1.63 and 9.95 mg AAE/g, 6.51 and 21.21 mg TE/g, and 2.02, and 15.58 mg AAE/g, respectively. Days to flowering, total fat, palmitic acid, oleic acid, and TPC were significantly influenced by origin and genotype differences, while seeds per pod, one-hundred seeds weight, CFC, DFC, vitamin C, RP, and TSC were not affected by these factors. Multivariate analysis categorized the accessions into four clusters showing significant variations in most of the analyzed parameters. Correlation analysis also revealed significant relationships between several noteworthy parameters. Overall, this comprehensive analysis of biochemical factors revealed diversity among the different yardlong bean varieties. These findings could have practical applications in industries, breeding programs, and conservation efforts.

## 1. Introduction

*Vigna unguiculata* (L.) Walp is one of the most important and popular legumes worldwide. In particular, two of its subspecies, including *V. unguiculata* (L.) Walp. ssp. *sesquipedalis* (yardlong bean) and *V. unguiculata* (L.) Walp. ssp. *unguiculata* (cowpea), are widely grown and consumed in various regions, including in Africa, Asia, North America, and Europe [1,2,3]. These subspecies are preferred for their ability to withstand heat and drought, making them a favorable choice for cultivation in the face of changing environments and climate conditions [4,5,6]. Additionally, they are recognized for their health benefits and nutritional values [5,7]. 

Yardlong bean, also called asparagus bean, is characterized by its elongated pods and is particularly popular in countries in Asia, including Korea, Thailand, Japan, and China [4,8,9]. In these countries, yardlong beans are often found in various dishes. The bean’s different parts, such as young pods, mature pods, green seeds, and ripe seeds, are used in different ways. They can be consumed raw, cooked with spices, or combined with other ingredients like vegetables, meat, or seafood [9,10,11]. 

Several studies have demonstrated that yardlong beans are rich in essential vitamins like vitamins A, C, and K, as well as important minerals such as potassium, iron, and manganese [12,13]. Moreover, the pods are low in calories, making them a nutritious option for those looking to maintain a healthy diet. Additionally, the beans are high in fiber, which not only aids in digestion but also helps in regulating blood sugar levels. Compared to other legumes, yardlong beans have a relatively low fat content, making them suitable for individuals following a weight management diet [14]. Yardlong beans are also a good source of protein, providing the body with essential amino acids [15,16,17]. Yardlong beans are not only rich in nutrients but also contain bioactive components such as phenolic acids, flavonoids, and saponins. These substances have diverse biological properties like antioxidant and anti-inflammatory effects [9,18]. Moreover, due to these beneficial metabolites, the regular consumption of yardlong beans has been associated with improved cardiovascular health, reduced cholesterol levels, as well as a decreased risk of chronic diseases such as cancer and diabetes [5,19,20]. The presence of these bioactive compounds in yardlong beans also suggests their potential applications in the pharmaceutical and nutraceutical industries. In summary, the rich biochemical profile of yardlong beans makes them a valuable addition to a healthy diet, offering a wide array of health benefits [9,12]. 

Despite having many benefits and desirable qualities, yardlong beans are still one of the most underutilized legumes [21]. Moreover, factors such as environmental conditions and genetic variances have a significant impact on the production and quality of yardlong beans [22,23]. Currently, genetic breeding methods are being used to improve the nutritional content, productivity, and disease resistance of various legumes, including those belonging to the genus *Vigna*. These initiatives also seek to tackle global food security issues and improve human health [12,24]. In this regard, understanding the diversity of plant genetic materials is essential to comprehend their overall phenotypic characteristics and set the foundation for future genomics research. Moreover, such information can help in identifying varieties with unique traits that can be utilized to develop cultivars with enhanced nutritional value and potential health benefits, ultimately supporting sustainable production. Unlike many other legumes, however, there is a scarcity of studies that have investigated a large collection of yardlong beans and their various phenotypic characteristics [6,15,17]. Therefore, this study aimed to assess the nutrient levels (crude fiber, dietary fiber, total fat, total protein, and vitamin C contents), fatty acids, secondary metabolite contents (total phenolic, total saponin, and total tannin contents), and antioxidant activities (DPPH^•^ scavenging activity, reducing power, and ABTS^•+^ scavenging activity) of 35 yardlong bean accessions originating from Korea, China, Myanmar, and Thailand along with their key agronomical traits. To the best of our knowledge, this study is among the first to statistically investigate how both origin and genotype differences affect these phenotypic characteristics. Overall, the results of this study may provide valuable insights into the diversity of yardlong beans in terms of their biochemical composition, laying the foundation for future genomic studies. 

## 2. Materials and Methods

### 2.1. Chemicals and Reagents

The chemicals and reagents utilized in this research were of analytical grade and used in their original state. Ethanol and sulfuric acid were bought from Fisher Scientific (Pittsburgh, PA, USA). The rest of the chemicals and reagents, including catechin, gallic acid, formic acid, L-ascorbic acid, anhydrous sodium carbonate (Na_2_CO_3_), Folin–Ciocalteu phenol reagent, potassium ferricyanide, trichloroacetic acid, vanillin, ferric chloride, 1,1-diphenyl-2-picrylhydrazyl (DPPH) radical, 2,2′-azino-bis(3-ethylbenzothiazoline-6-sulfonic acid) diammonium salt (ABTS), 6-hydroxy-2,5,7,8-tetramethylchroman-2-carboxylic acid, and fatty acid standards (palmitic acid, stearic acid, oleic acid, linoleic acid, and linolenic acid), were ordered from Sigma-Aldrich (St. Louis, MO, USA). 

### 2.2. Plant Materials Collection, Cultivation, and Preparation

The seeds of 35 yardlong bean varieties, consisting of 20 cultivars and 15 landraces, were sourced from the gene bank at the National Agro-biodiversity Center, Rural Development Administration (Jeonju, Republic of Korea). These yardlong beans were originally collected from Korea (*n* = 11), China (*n* = 18), Myanmar (*n* = 3), and Thailand (*n* = 3). For each accession, 25 seeds were sown on plug trays on 3 May 2022. After seventeen days, 9 randomly selected seedlings from each accession were transferred to an experimental farm located at the center (Jeonju, Republic of Korea) and planted in 80 cm apart rows. The seedling-to-seedling distance was kept at 30 cm. Field cultivation took place under uniform growth conditions and lasted until August 2022. The agronomic traits were recorded from field inspection and laboratory examination during seedling, vegetative, and maturity stages. Immature pods were harvested at a full green state, freeze-dried in an LP500 freeze dryer (ilShinBioBase in Dongducheon, Republic of Korea), powdered, and analyzed for their vitamin C content, total secondary metabolite contents, and antioxidant activities. Likewise, matured seeds were hand-harvested, freeze-dried, ground into powder, and analyzed for their nutritional components and fatty acid compositions. All powdered samples were stored at −20 °C when not in use. 

### 2.3. Determination of Nutritional Contents

Standard methods recommended by AOAC were utilized to analyze the nutritional components, including crude fiber, dietary fiber, total protein, and total fat, as detailed in a recently reported study [25,26]. The crude fiber content was assessed using a Fiber Analyzer (FOSS, Hillerod, Denmark) following the modified Henneberg and Stohmann approach. Likewise, the dietary fiber content was determined through an enzymatic–gravimetric assay using an Analytical Fibertec E-1023 System from FOSS. The Kjeldahl method was used to determine the total protein content, which was then calculated by multiplying the nitrogen content by 6.25. The total fat content was measured utilizing the soxhlet extraction technique with a Soxtec800 extractor from FOSS and n-hexane as the extracting solvent. Each analysis was conducted in triplicate, and the results are presented as percentages relative to the weight of the dried sample. 

### 2.4. Fatty Acid Analysis Using Gas Chromatography

Fatty acid methyl ester (FAME) derivatives were synthesized initially through a direct methylation method, as detailed previously without any modifications [26]. To analyze the fatty acids, a QP2010 gas chromatography–flame ionization detector instrument (Shimadzu, Kyoto, Japan) equipped with an HP-INNOWAX column (30 m × 0.250 mm, 0.25 µm) was employed. The sample injection volume was 1 µL with a split ratio of 50:1. Helium was served as a carrier gas at a flow rate of 1.5 mL/min. The detector and injection port temperatures were each set at 250 °C. During analysis, the column temperature was set at 100 °C and gradually increased to 170 °C at a rate of 60 °C/min with a 1 min hold time. Subsequently, the temperature was raised to 240 °C at a 6.5 °C/min ramp and maintained for an additional 1 min, totaling a 16.4 min run time. The LabSolution software version 5.92 (Shimadzu, Kyoto, Japan), was used to analyze the GC chromatograms acquired. The fatty acids were Identified by comparing retention times to their corresponding external standards (Appendix A), and their concentrations were calculated as peak area percentages. 

### 2.5. Vitamin C Analysis Using High-Performance Liquid Chromatography

Vitamin C was extracted using a solution of metaphosphoric acid (MPA) as outlined in a previous study with a few modifications [27]. Specifically, 1 g of powdered sample was placed in a 15 mL extraction tube and subsequently mixed with 10 mL of a solvent mixture containing 3% MPA and ethanol in an 8:2 ratio. The mixture was shaken for 45 min in an ice bath set at 4 °C in the dark, followed by centrifugation for 10 min (3134× *g*, 4 °C). The supernatant was then collected, filtered, and readied for injection. The concentration of vitamin C was determined using a 1260-Infinity Quaternary High-performance Liquid Chromatography system (Agilent Technologies, Santa Clara, CA, USA) with a Zorbax SB-18 column (4.6 × 250 mm, 5 µm) which was maintained at 30 °C. The sample injection volume was 2 µL, and the mobile phase consisted of a mixture of water with 0.1% trifluoroacetic acid (A) and methanol (B). The gradient condition began at 5% B, increased to 50% in 10 min, and then returned to 5% over another 10 min, with a post-run time of 5 min. The chromatogram was monitored at 245 nm through a diode-array-detector (DAD), and the content of vitamin C (in mg/g) was determined using *L*-ascorbic acid as a reference standard (Appendix A). 

### 2.6. Determination of Total Metabolite Contents

To measure the total secondary metabolite contents, samples were prepared using a procedure outlined in a previous study [28]. Initially, 0.5 g of powdered sample was mixed with 5 mL of 80% aqueous ethanol in a 15 mL extraction tube. The mixture was then subjected to ultrasonication in a water bath for 45 min at 25 °C. After centrifuging (3134× *g*, 10 min), the upper layer was preserved, and the remaining residue underwent a second round of extraction with 2.5 mL of the solvent. The combined liquid supernatant was used to analyze the total levels of secondary metabolites. The absorbance reading was recorded using an Eon Microplate Spectrophotometer (Bio-Tek, Winooski, VT, USA) for each test. The total phenolic (TPC), total tannin (TTC), and total saponin (TSC) contents were measured according to a recently published protocol [26]. TPC was determined using the Folin–Ciocalteu method and expressed as milligrams of gallic acid equivalents per gram of dried sample weight (mg GAE/g), with gallic acid used as a reference standard. The total tannin content (TTC) was evaluated using the vanillin–HCl method, with catechin as a standard for calibration curves. TTC was recorded as milligrams of catechin equivalents per gram of dried sample weight (mg CE/g). The vanillin–sulfuric acid test was used to measure the total saponin content (TSC), with diosgenin employed to create calibration curves. TSC was then calculated as milligrams of diosgenin equivalent per gram of dried sample (mg DE/g). 

### 2.7. Determination of Antioxidant Activities

The samples used to analyze the total amount of secondary metabolites were also used to assess antioxidant activities, which were measured using in vitro colorimetric tests, as described in a previously reported study [26]. The DPPH^•^ scavenging activity and the reducing power were reported in milligrams of ascorbic acid equivalents per gram of dried sample weight (mg AAE/g), with ascorbic acid as the reference standard. Trolox was employed as a standard for ABTS^•+^ scavenging activity, and the results were recorded in milligrams of Trolox equivalents per gram of dried sample weight (mg TE/g) using the following formula (Equation (1)).
(1)$${ABTS}^{•+}\mathrm{scavenging}\;\mathrm{activity}\left(mg\;TE\;g^{-1}DW\right)=\frac{\left(C\times V\times Df\right)}{m}$$
where C is the Trolox concentration (mg/mL) corresponding to the calibration curve of the sample, V is the sample volume (mL), Df is the dilution factor, and m is the sample weight (g).

### 2.8. Statistical Analysis

In this study, all measurements and analyses were carried out in triplicate, unless specified, and the results are presented as the mean ± standard deviation (SD). Statistical analysis was performed using an analysis of variance (ANOVA), followed by Duncan’s multiple range test with xlstat software-2019.2.2 (Lumivero, CO, USA). Box plots and Pearson’s correlation matrix were computed using R-software (version 4.0.2, r-project, www.r-project.org). For hierarchical cluster analysis and principal component analysis, JMP-Pro software version-17 (SAS, Inc., Cary, NC, USA) was utilized. 

## 3. Results and Discussion

### 3.1. Field Performances and Characteristics of Yardlong Beans

The 35 yardlong bean accessions were characterized by their key agronomic performances, including nine qualitative traits and six quantitative traits. Details regarding each of the agronomical performances of the accessions can be found in Appendix A, along with their introduction and/or temporary number (Appendix A). In summary, the yardlong bean accessions did not show wide differences in their qualitative agronomic performances. Despite this, the presence of several states in pod color and seed color, which were scaled based on visual observation, was noteworthy and agreed with previous studies [4,29]. Regarding quantitative traits, days to flowering (DF) and days to maturity (DM) were in the ranges of 47–59 days and 69–88 days, with coefficients of variation (CVs) of 5.76 and 17.78%, respectively. Likewise, yield traits such as pod length (PL), number of seeds per pod (SPP), and one-hundred seeds weight (HSW) were in the ranges of 23.80–71.40 cm, 11.20–19.00, and 8.10–22.23 g, respectively. These observed ranges were comparable with many previous studies [15,29,30]. Early maturing and high-yielding legumes are of great interest in the breeding program [17,31]. In this study, 22 accessions had lower DM than the total average (75.63 days). Similarly, 18 accessions had higher PL, 21 accessions had higher SPP, and 17 successions had higher HSW than the average values. Among these, early maturing cultivars Tichun zhi jiang 28-2, Te xuan zhang tang wang, and landrace San chi lu, all from China, simultaneously displayed higher SPP, PL, and HSW than the average values. Therefore, those accessions with desirable properties in these regards could be valuable resources [17]. 

### 3.2. Nutritional Components

The nutritional compositions, including crude fiber (CFC), dietary fiber (DFC), total protein, and total fat contents, of the 35 yardlong bean accessions are provided in Figure 1. The CFC and DFC levels ranged from 4.10 to 6.51% and from 16.71 to 23.49%, respectively, with CVs of 11.07% and 7.46% (Figure 1a). Similarly, the total protein content (Figure 1b) ranged from 22.45 to 28.11% (CV: 4.21%), and the total fat content ranged from 0.59 to 2.00% (CV: 28.14%) (Figure 1c). The ranges observed in this study were comparable to those found in previous studies. For instance, Bai et al. [32] found DFC, total protein, and total fat contents in the ranges of 29.49–38.54%, 17.30–27.23%, and 1.87–3.14%, respectively, in 24 Chinese cultivars. Onwuliri and Obu [33] also reported a total protein content ranging from 20.5 to 39.7%, a total fat content ranging from 1.14 to 3.03%, and a CFC ranging from 1.70 to 4.50% in six cultivars grown in Nigeria. Several other studies also reported comparable ranges, indicating consistency with the findings observed in this study [34,35,36]. Statistical analysis showed significant variations among the yardlong beans (*p* < 0.05). Tanjingeun, a Korean landrace, simultaneously displayed the highest CFC and DFC, with the former being significantly different from the other varieties, while the latter was significantly different from all others except for Techang jinqili, 901 Zaoshu Jiangdou, Techang 902 jiangdou, and Gangwonpyeongchang-2003-4, which had the second (21.74%), third (21.65%), fourth (21.46%), and fifth (21.20%) highest DFC levels, respectively. The lowest CFC was found in SD 3135, another Korean landrace, and the lowest DFC in Tichun zhi jiang 28-2, a Chinese cultivar. Regarding protein and fat contents, the cultivar THA-JSH-2008-81021 from Thailand had the highest, which was significantly different from the other accessions (*p* < 0.05). In comparison, the landrace Hei Mei 1 from China showed the highest total fat content, which was significantly different from the other accessions, except for the cultivar Chunqiu Hong Jiangdou from China, which had the second highest TF at 1.94%. On the other hand, the cultivar Man Di Hong Wu Jia Dou from China had the lowest TP, while the cultivar Yardlong Bean 287/2556 from Thailand had the lowest TF. Notably, the cultivar Tanjingeun, which had the highest CFC and DFC, had the second-lowest total fat at 0.69% and the fifth-highest total protein at 26.89%. The consumption of yardlong beans which are high in fiber is linked to a reduced level of bad cholesterol and risk of cardiovascular diseases and cancer, as highlighted before [7,12]. Furthermore, yardlong beans that are rich in protein and low in fat are desirable, making them excellent amino acid sources [7]. In this study, 18 accessions showed CFC above the average (4.92%), 15 had higher DFC than the average (19.30%), another 15 had higher total protein content than the average (25.77%), and 21 had a lower total fat value than the average (1.23%). Among these, two Chinese cultivars (Taiwan chunqiu hong and Guamian hong jiangdou), one Korean landrace (Tanjingeun), and one landrace from Myanmar (Tai Htaung Pe Ni) simultaneously exhibited higher total protein content, CFC, and DFC than the average values. Interestingly, all except Taiwan chunqiu hong also had lower total fat contents than the average, suggesting that these accessions could be valuable sources of healthy nutrition [12].

### 3.3. Fatty Acid Contents

The fatty acid composition and contents of each yardlong bean variety were analyzed using GC-FID. As shown in Table 1, all five major fatty acids were detected in each accession with significantly different levels (*p* < 0.05). The CVs ranged from 3.36% for linoleic acid to 16.06% for oleic acid. The levels of palmitic acid and stearic acid, which are both saturated fatty acids, ranged from 29.08% to 32.45% and 3.4% to 5.29%, respectively. Similarly, the unsaturated fatty acids, including oleic acid, linoleic acid, and linolenic acid, had levels ranging from 7.88 to 16.36%, 34.39 to 39.26%, and 14.41 to 18.12%, respectively. Compared to other legumes, *V. unguiculata* subspecies have low levels of fats, but they contain fatty acids with varying degrees of saturation and oxidation [7,34]. The analysis of fatty acids in this study identified palmitic acid as the primary saturated fatty acid and linoleic acid as the dominant unsaturated fatty acid in all the yardlong accessions. These observations were consistent with previous findings [32,37]. Statistical analysis showed that Taiwan chunqiu hong from China and Gangwonpyeongchang-2003-4 from Korea equally had the highest palmitic acid, while Tichun zhi jiang 28-2 from China had the lowest (*p* < 0.05). Likewise, Gaotian jinguan from China had the highest stearic acid, while the Nan 1 Variety from Thailand had the lowest (*p* < 0.05). Regarding unsaturated fatty acids, Te xuan zhang tang, Yard long Bean 287/2556, and Gyeongnamhabcheon-2019–2 had the highest oleic acid, linoleic acid, and linolenic acid contents, respectively, with the first two being cultivars from China and Thailand, and the last one being a landrace from Korea. Tanjingeun also had the lowest oleic acid content, while Qiu jiang 512 and Techang 902 jiangdou from China had the lowest linoleic acid and linolenic acid contents, respectively (*p* < 0.05). In terms of total fatty acid content (Figure 2), PUFA exceeded TSFA and MUFA in all of the yardlong bean varieties, which again corroborates with previous observations [32,37]. The presence of high levels of unsaturated fatty acids makes yardlong beans a good source of fats rich in omega-3 and omega-6 [7,38]. However, high levels of PUFA may reduce shelf life due to the oxidative nature of these fatty acids [39]. In this study, the yardlong bean accessions were found to be unfavorable in this aspect, as indicated by the double bond index, which ranged from 131.11 to 138.49. On the other hand, the ratio of TUFA to TSFA ranged from 1.73 in Taiwan chunqiu hong to 2.03 in Nan 1 Variety, all having ratios above 0.45. This suggests that the yardlong bean accessions have the potential to be a valuable source of oil rich in fatty acids for the prevention of cardiovascular ailments [37,39].

### 3.4. Vitamin C Content

The level of vitamin C in each of the yardlong bean accessions was analyzed using HPLC, and a significant variation was observed among the accessions (*p* < 0.05). The content of vitamin C ranged from 0.23 mg/g in the San chi lu landrace from China to 3.04 mg/g in landrace Chungbuk Geosan 2011-18 from Korea, showing an over 10-fold variation among the yardlong beans (Figure 1d). The vitamin C content in landrace Chungbuk Geosan 2011-18 was significantly higher compared to other accessions, except for Hei mei huang zi wang and Techang 908 jiangdou, which had the second (2.91 mg/g) and third (2.87 mg/g) highest levels, respectively (*p* < 0.05). Though yardlong beans are generally considered to be rich in vitamin C compared to other legumes, there has been a lack of thorough analysis across a wide array of genetic materials [14]. Previous studies found higher levels of vitamin C in the leaves than in the seeds and pods of *V. unguiculata* subspecies, recommending sprouting the seeds to boost the vitamin C content [40,41,42,43]. Vitamin C plays a crucial role in supporting the immune system and maintaining skin health, among others [12,43]. This study discovered 17 accessions that had vitamin C levels exceeding the average (2.09 mg/g). Out of these, landraces Gatggeungkong and Chungbuk Geosan 2011-18, along with cultivar Gangwonpyeongchang-2003-17 from Korea, exhibited higher total protein and dietary fiber contents and lower fat content than the average levels, making them valuable sources of healthy nutrition. 

### 3.5. Total Secondary Metabolite Contents 

In addition to their nutrient levels, the yardlong beans were found to have high levels of polyphenols, with significant differences in the total phenolic, tannin, and saponin contents. Table 2 presents the total metabolite contents of each yardlong bean accession. The TPC, TTC, and TSC ranged from 3.78 to 9.13 mg GAE/g, 31.20 to 778.34 mg CE/g, and 25.79 to 82.55 mg DE/g, respectively, showing approximately 2.4-fold, 25-fold, and 3.2-fold variations. The Chinese cultivar Man di hong wu jia dou exhibited the highest TPC and TSC, with the former being significantly different from other accessions and the latter significantly different from all accessions except Guamian hong jiangdou, Taiwan chunqiu hong, and Tai Htaung Pe Ni, which had the second (81.01 mg DE/g), third (76.06 mg DE/g), and fourth (75.77 mg DE/g) highest TSC levels (*p* < 0.05). Similarly, Taiwan chunqiu hong, another Chinese cultivar, had the highest TTC, which was significantly different from other accessions. The lowest TPC, TTC, and TSC were found in landraces Chungbuk Geosan 2011-245 from Korea, Sung 99 from Myanmar, and Gyeongnamhabcheon-2019-2 from Korea, respectively. Compared to cowpeas, the level of secondary metabolites and antioxidant properties of yardlong beans has hardly been investigated [35,44,45]. Moreover, there were differences in the plant parts studied, the methods of analysis, and how the findings were reported, making it challenging to draw direct comparisons. Polyphenols, in general, are known for their capability to neutralize harmful free radicals in the human body, making plant varieties with high levels of these substances valuable resources [28,46,47]. In contrast, tannins found in legumes are regarded as anti-nutrient factors and may hinder protein digestion, leading to a preference for lower concentrations [21]. In this regard, accessions like Sung 99, Te xuan zhang tang wang, Tichun zhi jiang28-2, Hang xin taikong wu jia dou, and Qiu jiang 512, which had much lower total tannin contents (<40.00 mg DE/g), could be important resources [48].

### 3.6. Antioxidant Activities 

Various in vitro assays are commonly utilized to evaluate the antioxidant capacities of dietary crops, each presenting distinct radical scavenging mechanisms. Therefore, conducting multiple assays is essential as a single assay may not offer a comprehensive assessment of antioxidant properties [47]. In this research, three separate assays were employed, revealing significant variations in antioxidant activities between the yardlong bean accessions (Table 2). The DPPH^•^ scavenging activity, ABTS^•+^ scavenging activity, and RP were in the ranges of 1.63–9.95 mg AAE/g, 6.51–21.21 mg TE/g, and 2.02–15.58 mg AAE/g, respectively, showing approximately 4-fold, 3-fold, and 8-fold variances. The variations in values between the assays could be attributed to their different mechanisms of action [47]. Previous studies also showed wide-ranging antioxidant activities in various *V. unguiculata* species, although there were discrepancies in the antioxidant assay, sample type, and reporting method [35,44,45]. Taiwan chunqiu hong displayed the highest DPPH^•^ scavenging activity and RP simultaneously, each being significantly different from the rest of the accessions, except for Man di hong wu jia dou, which showed the second highest levels of each (9.27 and 14.69 mg AAE/g, respectively). Taiwan chunqiu hong also displayed the second highest ABTS^•+^ scavenging activity (18.87 mg TE/g). As described before, this cultivar had the highest TTC, the second highest TPC, and the third highest TSC. Man di hong wu jia dou, which displayed the highest TPC and TSC, had the highest ABTS^•+^ scavenging activity, which was significantly different from the rest of the accessions (*p* < 0.05). The lowest DPPH^•^ scavenging activity, ABTS^•+^ scavenging activity, and RP were found in Gyeongnamhabcheon-2019-2, 901 Zaoshu jiangdou, and Te xuan zhang tang wang, respectively. Generally, accessions with higher polyphenol contents demonstrated higher antioxidant properties, while those with lower levels exhibited the opposite trend (Table 2). Similar findings have been documented in various legumes such as soybeans, common beans, mung beans, chickpeas, and lentils, once again underscoring the importance of polyphenols in neutralizing reactive radicals in the human body [49,50].

### 3.7. Effects of Genotype and Origin Differences 

The levels of agronomic traits and biochemical components showed wide variability in this study, possibly due to genetic differences among the yardlong bean accessions. Additionally, disparities in growing conditions, cultivation year, and the number of genotypes examined could have influenced the differences observed compared to previous studies. Overall, studying the impact of environmental and genetic factors on the biochemical characteristics and agricultural traits of crops is crucial for choosing superior varieties and carrying out genomic studies [51]. In this study, statistical analysis was also conducted to examine the impacts of genotype and origin variations on the parameters assessed (Table 3). 

In terms of agronomic traits, cultivars generally outperformed landraces in most of the parameters, with all except for DF and DM being higher in the cultivars. However, statistical analysis showed that none of the agronomic traits, except for DF, were significantly affected by genotype differences. In contrast, the difference in origin significantly affected all except for HSW and SPP (Table 3). Specifically, Chinese accessions were found to be early flowering and maturing, with average DF and DM of 52.56 and 73.78 days, respectively (*p* < 0.05). Likewise, Chinese accessions had the highest average SPP (16.14) and pod length (55.36 cm) and the lowest average HSW (15.59 g). Overall, the observed results signified that origin might be a good parameter to assess yardlong beans according to their maturity and pod-related yield components.

Concerning nutritional components, only total protein and vitamin C contents were higher in landraces, while the rest, including total fat, CFC, and DFC, were higher in cultivars. However, all variations were not significant, except for total fat content. Based on origin, Chinese accessions had the highest average total fat content, which was significantly different from other origins (*p* < 0.05). Additionally, Chinese accessions also had the highest average CFC and the second-highest average DFC, but they had the lowest average total protein content. In contrast, Thailand accessions had the highest average total protein content and DFC, while Korean accessions had the lowest average total fat content, CFC, and DFC. Nevertheless, these differences, except for total fat content, were not statistically significant. Regarding vitamin C content, the landraces displayed the highest average content (2.11 mg/g) compared to the cultivars (2.08 mg/g), although the difference between them was not statistically significant (*p* < 0.05). When considering origin, the average vitamin C content ranked in the order of Korea (2.32 mg/g) > China (2.01 mg/g) > Thailand (1.94 mg/g) > Myanmar (1.93 mg/g). Once again, the difference in origin did not have a significant effect on the level of vitamin C. The overall findings signify that neither origin nor genotype significantly affects the levels of nutritional components, and, hence, individual genotypes should be assessed to determine their overall nutritional values [32,36,49]. 

Unlike the nutritional factors discussed before, most of the fatty acid contents were significantly influenced by both genotype and origin differences. Landraces typically showed higher average contents of palmitic acid and linolenic acid, while cultivars had higher levels of stearic acid, oleic acid, and linoleic acid contents (Table 3). Interestingly, significant differences were found in all fatty acids except for stearic acid and linoleic acid (*p* < 0.05). Generally, landraces had a notably higher level of TSFA, whereas cultivars had the highest concentration of TUFA (*p* < 0.05). Origin difference also affected the fatty acid composition, showing significant variations in all the fatty acids except for linoleic acid (*p* < 0.05). Korean and Chinese accessions had the highest average levels of palmitic acid and stearic acid, respectively, while Thailand accessions had the lowest levels of both and the highest linoleic acid level. Chinese varieties had the highest levels of oleic acid and the lowest levels of linoleic acid and linolenic acid, while Myanmar accessions had the lowest levels of oleic acid and the highest levels of linolenic acid. Overall, Korean varieties were characterized by high TSFA levels, followed by Myanmar, China, and Thailand, whereas the opposite trend was observed in terms of the TUFA level (Table 3). These results highlighted that both genotype and origin could be ideal parameters in discriminating a large population of yardlong beans according to their fatty acid levels [51].

When considering total secondary metabolite contents and antioxidant activities, cultivars once again showed superiority over landraces, with significant differences observed in TPC and DPPH^•^ scavenging activity (*p* < 0.05). Concerning origin, Thailand accessions had the highest average levels of TPC, TTC, and all three antioxidant activities, with the former and ABTS^•+^ scavenging activity showing significant variations (*p* < 0.05). Likewise, Myanmar accessions displayed the highest average TSC level. In contrast, Korean accessions displayed the lowest levels of all the total secondary metabolite contents and antioxidant activities. Previously, several studies investigated the impacts of genotype difference and origin on different biochemical factors in other legumes and cereals [52,53,54,55,56]. In yardlong beans, however, such studies are scarce. The results of this study demonstrated that various traits such as DF, total fat content, TPC, specific fatty acids such as palmitic acid and oleic acid, TSFA, and TUFA were significantly affected by the differences in origin and genotype. As a result, these factors can be targeted for differentiation among different yardlong bean genotypes [55].

### 3.8. Multivariate Analysis

#### 3.8.1. Hierarchical Cluster (HCA) and Principal Component (PCA) Analyses

Multivariate statistical tools such as HCA and PCA are of great importance in viewing the relative distribution of plant genetic materials and their relationship with phenotypic data [34,53,54]. In this study, the HCA classified the 35 yardlong bean accessions into four major clusters, revealing significant differences in most of the parameters between them. The majority of the accessions were grouped in cluster I, which contained 14 accessions, followed by cluster IV, which contained 13 accessions (Figure 3). 

Clusters II and III each contained four accessions. Statistical analysis showed that all the quantitative traits except for SPP, HSW, CFC, and DFC showed significant variations (*p* < 0.05) among the clusters (Figure 4). Specifically, the accessions in cluster II were late to flowering and maturing, with average DF and DM of 51.69 and 72.39 days, respectively, compared to the other clusters. Despite these, they were characterized by having the highest average total protein, total fat, CFC, DFC, TSC, TTC, DPPH^•^ scavenging activity, RP, palmitic acid, and TSFA, and the lowest average TUFA, linolenic acid, and oleic acid. They also displayed the second highest average vitamin C content (2.11 mg/g) next to cluster I (2.24 mg/g), TPC (97.13 mg GAE/g) and ABTS^•+^ scavenging activity (16.21 mg TE/g) next to cluster III (7.55 mg GAE/g and 17.14 mg TE/g, respectively), and stearic acid (4.28%) next to cluster IV (4.68%). Overall, the HCA result signified that the yardlong bean accessions were grouped according to their relative performances. 

PCA was also computed to further view the distribution of the yardlong bean accessions and their association with the analyzed parameters (Figure 5). In the PCA, a total of seven components with eigenvalues greater than one, which together contributed over 85% of the total variance, were obtained (Appendix A). Out of these, the first two components (PC1 and PC2) accounted for 48.14%, showing their superior contribution to the total variance, and were used to compute the overall PCA. As can be seen in the score plot (Figure 5a), the accessions were grouped according to their cluster, supporting the HCA observation. Origin-based and genotype-based PCA also clustered the Korean landraces compared to the others. The loading plot (Figure 5b) showed DM, TSC, TTC, ABTS^•+^ scavenging activity, reducing power, linoleic acid, and PUFA as the major discriminating factors along PC1, with factor loadings of ≥±0.50 and contributions ranging from 4.21 to 9.03% (Appendix A). Likewise, total fat (9.92%), TSFA (10.61%), and TUFA (10.61%) were the major contributors along PC2. Interestingly, days to flowering, TPC, DPPH^•^ scavenging activity, palmitic acid, and oleic acid had factor loadings ≥ ±0.50 across both PC1 and PC2, making them the most important discriminating factors (4.21–10.71%). It is important to note that these traits were significantly affected by variations in both genotype and origin. The grouping of the variables along PC1 and PC2 was also in agreement with the observation in the HCA (Figure 5b). Overall, the findings from the PCA were consistent with those of the HCA, indicating the importance of agricultural characteristics and biochemical factors in classifying a wide variety of yardlong beans [31,53,54].

#### 3.8.2. Pearson’s Correlation Analysis

Pearson’s correlation analysis was computed to quantitatively determine the association between the analyzed parameters (Figure 6). The correlation analysis revealed significant and notable associations among the different parameters. Among the agronomic traits, DM and HSW showed a positive association with each other (*r* = 0.41, *p* < 0.05), as well as with TTC (*r* = 0.65, *p* < 0.001 and 0.47, *p* < 0.01, respectively) and total protein content (*r* = 0.49, *p* < 0.01, and *r* = 0.57, *p* < 0.001, respectively). These observations signified that the maturity period could affect the biochemical components in yardlong beans [57]. Likewise, the weak and/or negative correlation of vitamin C with the rest of the biochemical parameters was notable, which could be associated with its distinct biosynthetic pathways [58]. The secondary metabolites contents, including TPC, TTC, and TSC, showed strong and positive correlations with all of the antioxidant activities, including DPPH^•^ scavenging activity (*r* = 0.81, 0.78, and 0.78, respectively), ABTS^•+^ scavenging activity (*r* = 0.86, 0.78, and 0.77, respectively), and RP (*r* = 0.82, 0.80, and 0.83, respectively), each being significant at *p* < 0.001. Several studies also noted similar associations, further confirming their roles in controlling reactive radicals in the human body [16,26,28,45,50]. Among the fatty acids, the positive and significant relationships between oleic acid and TUFA (*r* = 0.70, *p* < 0.001), as well as palmitic acid with TSFA (*r* = 0.93, *p* < 0.001), indicate their dominance within their respective groups. This, in turn, resulted in a negative and strong correlation between TUFA and TSFA (*r* = −1.00, *p* < 0.001). Oleic acid also showed negative associations with the other unsaturated fatty acids, including linoleic acid (−0.68, *p* < 0.001) and linolenic acid (*r* = −0.33), highlighting the role of desaturase enzymes in interconverting these fatty acids, a common process in fatty acid biosynthesis in legumes and crops [59]. Unlike the other fatty acids, linoleic acid also showed strong and positive associations with total secondary metabolites and antioxidant activities (0.58 ≤ *r* ≤ 0.71, *p* < 0.001). Previous studies on different legumes also reported comparable results, indicating the role of polyunsaturated fatty acids as antioxidants [60]. Other detailed and notable association levels between agronomic traits, biochemical contents, and antioxidant activities can be viewed in Figure 6.

## 4. Conclusions

Studying various characteristics in crop genetic materials, such as agronomic traits, nutrient levels, secondary metabolites, and biological activities, is crucial for understanding the relationship between phenotypic parameters and selecting the most appropriate varieties for consumption, breeding, and distribution to different stakeholders. In this research, 35 yardlong beans obtained from China, Korea, Myanmar, and Thailand were explored, focusing on their diverse biochemical compositions, antioxidant properties, and key agronomic traits. The results showed significant variations in each parameter, possibly due to genetic diversity within the yardlong bean accessions. Statistical analysis indicated that origin and genotype influenced certain biochemical components and agronomic traits, suggesting their potential usefulness in distinguishing among large collections of genetic materials. Furthermore, the study pinpointed specific yardlong bean accessions that excelled in terms of nutrient levels, secondary metabolite contents, and antioxidant activities, making them appropriate for dissemination across farms, use in breeding programs, and application in the food industry. The diversity observed among the yardlong beans also presents an excellent opportunity for future research in metabolomics, the investigation of anti-nutrient factors, and genomics to explore their genetic makeup and enhance their uses in breeding.

## Figures and Tables

**Figure 1 antioxidants-13-00463-f001:**
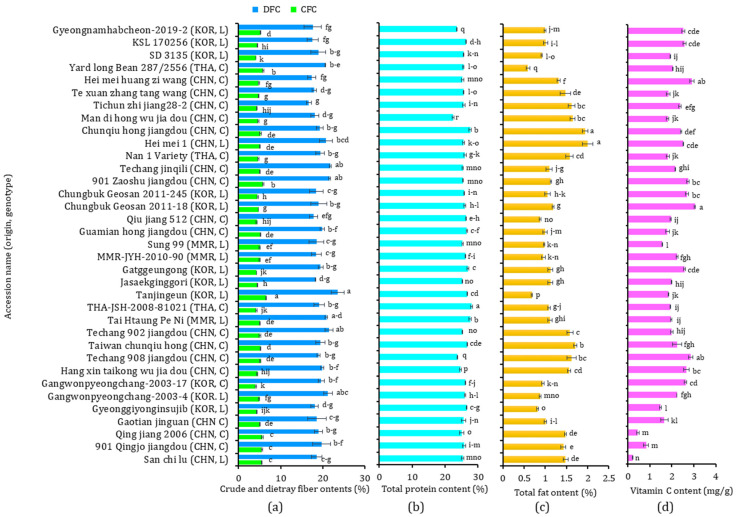
Crude fiber and dietary fiber contents (**a**), total protein content (**b**), total fat content (**c**), and vitamin C content (**d**) in 35 yardlong bean accessions grown in Korea. Different letters on bars in a category indicate significantly different mean values between the yardlong bean accessions (*p* < 0.05). C: cultivar, CFC: crude fiber content, CHN: China, DFC: dietary fiber content; KOR: Korea, L: landrace, MMR: Myanmar, TF: total fat, THA: Thailand, TP: total protein.

**Figure 2 antioxidants-13-00463-f002:**
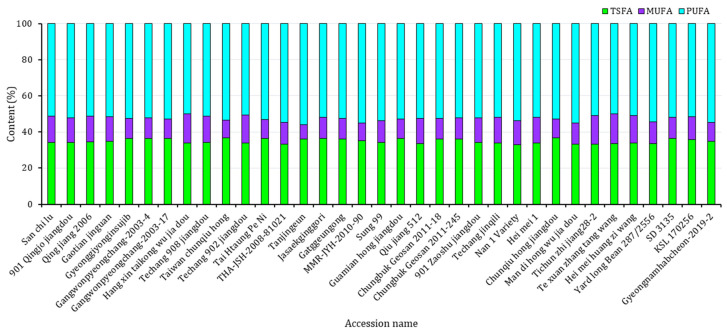
Contents of total fatty acids according to their degree of unsaturation across 35 yardlong bean accessions grown in Korea. MUFA: total monounsaturated fatty acid, PUFA: total polyunsaturated fatty acid, TSFA: total saturated fatty acid.

**Figure 3 antioxidants-13-00463-f003:**
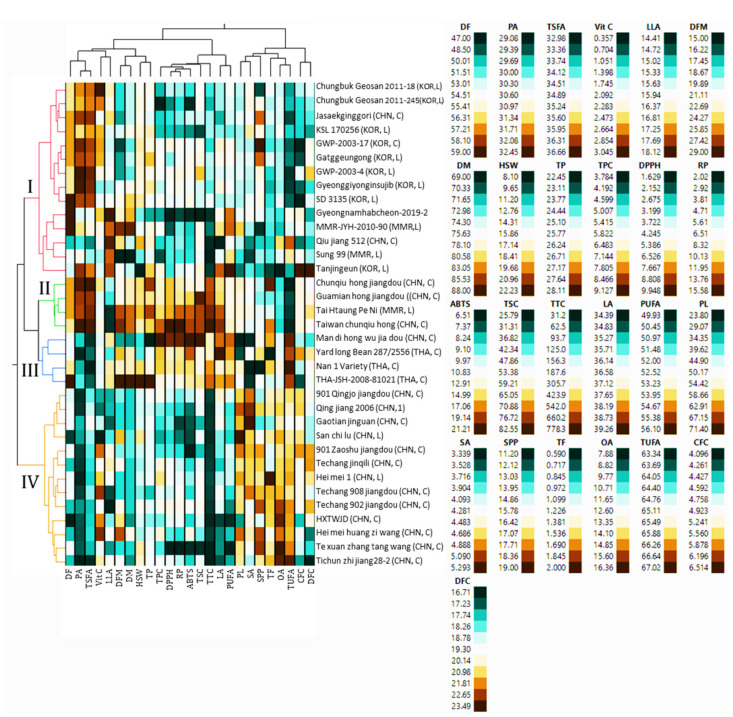
Two-way hierarchical cluster analysis matrix. ABTS: ABTS^•+^ scavenging activity, C: cultivar, CFC: crude fiber content, CHN, China, DF: days to flowering, DFC: dietary fiber content, DFM: days from flowering to maturity, DM: days to maturity, DPPH: DPPH^•^ scavenging activity, GWP-2003-4: Gangwonpyeongchang-2003-4, GWP-2003-7: Gangwonpyeongchang-2003-17, HSW: one-hundred seeds weight, HXTWJD: Hang xin taikong wu jia dou, KOR: Korea, L: landrace, LA: linoleic acid, LLA: linolenic acid, MMR: Myanmar, OA: oleic acid, PA: palmitic acid, PL: pod length, PUAF: total polyunsaturated fatty acid, RP: reducing power, SA: stearic acid, SPP; number of seeds per pod, TF: total fat, THA: Thailand, TP: total protein, TSFA: total saturated fatty acid, TUFA: total unsaturated fatty acid, TPC: total phenolic content, TSC: total saponin content, TTC: total tannin content, Vit C: vitamin C.

**Figure 4 antioxidants-13-00463-f004:**
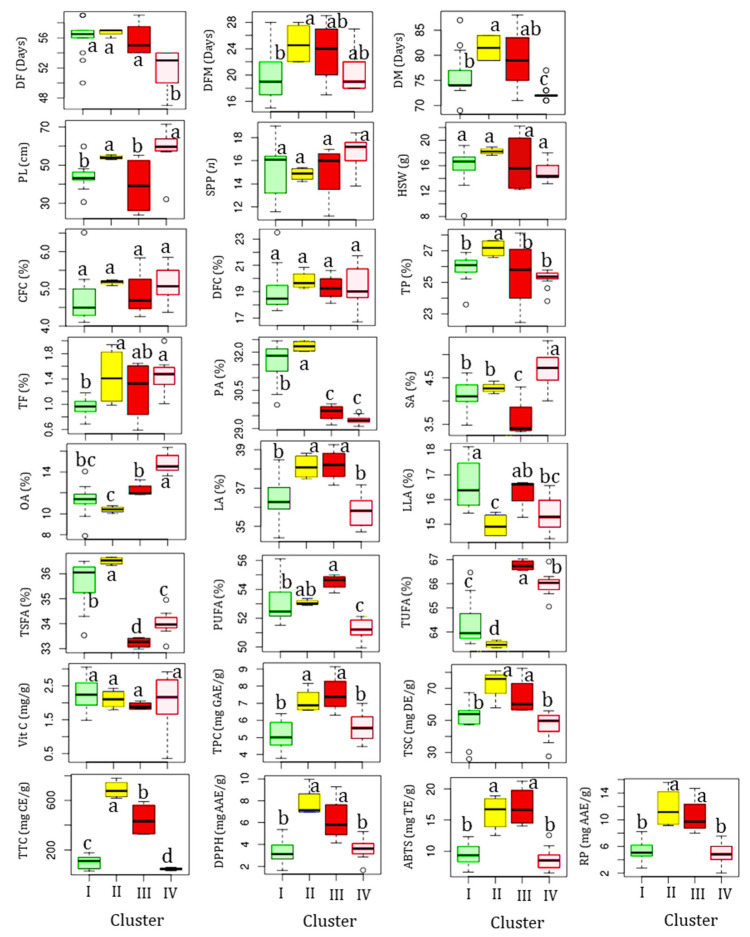
Boxplots showing variations in the analyzed parameters across clusters. Different superscript letters on boxplots in a category indicate significantly different means (*p* < 0.05). The descriptions of all abbreviations are similar to those stated in Figure 3.

**Figure 5 antioxidants-13-00463-f005:**
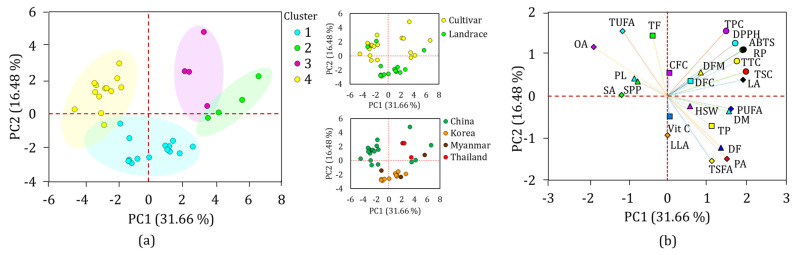
Score plot of yardlong bean accessions (**a**) and loading plot of variables (**b**) obtained from PCA. ABTS: ABTS^•+^ scavenging activity, CFC: crude fiber content, DF: days to flowering, DFC: dietary fiber content, DFM: days from flowering to maturity, DM: days to maturity, DPPH: DPPH^•^ scavenging activity, HSW: one-hundred seeds weight, LA: linoleic acid, LLA: linolenic acid, OA: oleic acid, PA: palmitic acid, PL: pod length, PUAF: total polyunsaturated fatty acid, RP: reducing power, SA: stearic acid, SPP; number of seeds per pod, TF: total fat, TP: total protein, TSFA: total saturated fatty acid, TUFA: total unsaturated fatty acid, TPC: total phenolic content, TSC: total saponin content, TTC: total tannin content, Vit C: vitamin C.

**Figure 6 antioxidants-13-00463-f006:**
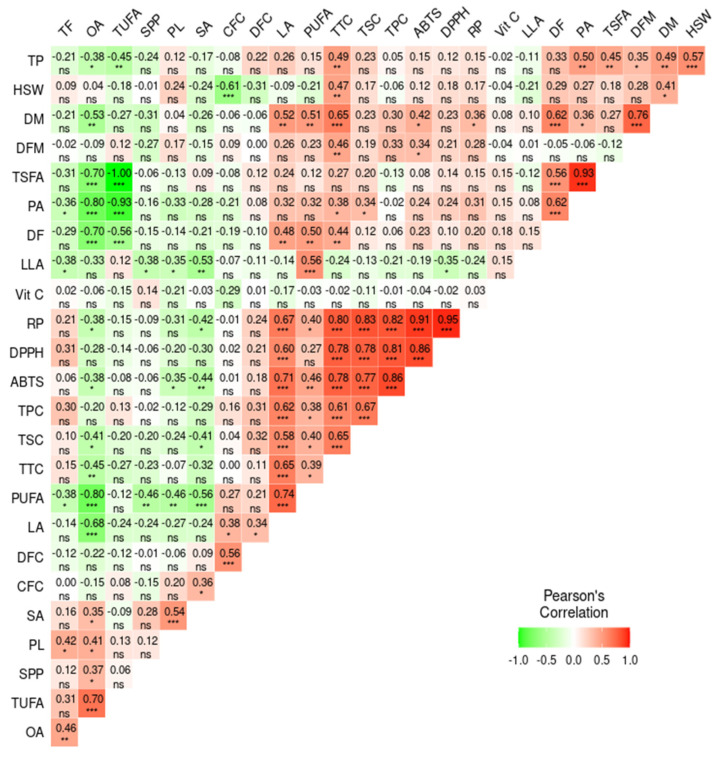
Pearson’s correlation matrix showing the association between the analyzed quantitative parameters. *** *p* < 0.001, ** *p* < 0.01, * *p* < 0.05, ns: not significant. ABTS: ABTS^•+^ scavenging activity, CFC: crude fiber content, DF: days to flowering, DFC: dietary fiber content, DFM: days from flowering to maturity, DM: days to maturity, DPPH: DPPH^•^ scavenging activity, HSW: one-hundred seeds weight, LA: linoleic acid, LLA: linolenic acid, OA: oleic acid, PA: palmitic acid, PL: pod length, PUAF: total polyunsaturated fatty acid, RP: reducing power, SA: stearic acid, SPP; number of seeds per pod, TF: total fat, TP: total protein, TSFA: total saturated fatty acid, TUFA: total unsaturated fatty acid, TPC: total phenolic content, TSC: total saponin content, TTC: total tannin content, Vit C: vitamin C.

**Table 1 antioxidants-13-00463-t001:** Variations in individual fatty acid contents across 35 yardlong bean accessions cultivated in Korea.

Accession Name	Individual Fatty Acid Contents (%)					DBI	TUFA: TSFA
	Palmitic Acid	Stearic Acid	Oleic Acid	Linoleic Acid	Linolenic Acid		
San chi lu	29.58 ± 0.04 ^o^	4.72 ± 0.02 ^l^	14.59 ± 0.03 ^kl^	35.81 ± 0.02 ^a^	15.29 ± 0.04 ^ijk^	132.10	1.92
901 Qingjo jiangdou	29.20 ± 0.04 ^mn^	4.95 ± 0.02 ^abc^	13.74 ± 0.03 ^ef^	37.16 ± 0.12 ^c–h^	14.95 ± 0.12 ^m^	132.92	1.93
Qing jiang 2006	29.26 ± 0.05 ^mn^	5.16 ± 0.02 ^ab^	14.24 ± 0.08 ^de^	36.34 ± 0.06 ^f–l^	15.00 ± 0.08 ^m^	131.92	1.91
Gaotian jinguan	29.66 ± 0.04 ^ij^	5.29 ± 0.02 ^a^	13.62 ± 0.02 ^ef^	36.68 ± 0.04 ^d–k^	14.74 ± 0.04 ^mno^	131.22	1.86
Gyeonggiyonginsujib	32.42 ± 0.10 ^a^	4.01 ± 0.01 ^ghi^	11.09 ± 0.07 ^h–j^	37.03 ± 0.07 ^c–i^	15.45 ± 0.18 ^jkl^	131.51	1.75
Gangwonpyeongchang-2003-4	32.45 ± 0.01 ^a^	4.05 ± 0.02 ^ghi^	11.32 ± 0.02 ^g–j^	36.20 ± 0.06 ^f–l^	15.99 ± 0.08 ^fg^	131.69	1.74
Gangwonpyeongchang-2003-17	32.26 ± 0.08 ^abc^	4.01 ± 0.01 ^ghi^	10.95 ± 0.12 ^h–l^	37.18 ± 0.02 ^c–h^	15.60 ± 0.06 ^ij^	132.10	1.76
Hang xin taikong wu jia dou	29.39 ± 0.10 ^klm^	4.45 ± 0.00 ^d–g^	16.05 ± 0.08 ^ab^	34.71 ± 0.05 ^kl^	15.40 ± 0.08 ^jkl^	131.69	1.96
Techang 908 jiangdou	29.31 ± 0.01 ^lmn^	4.94 ± 0.00 ^abc^	14.54 ± 0.04 ^cde^	36.54 ± 0.07 ^e–k^	14.68 ± 0.08 ^nop^	131.64	1.92
Taiwan chunqiu hong	32.45 ± 0.03 ^a^	4.16 ± 0.02 ^fgh^	10.03 ± 0.04 ^jkl^	38.83 ± 0.12 ^bc^	14.53 ± 0.16 ^op^	131.28	1.73
Techang 902 jiangdou	29.55 ± 0.03 ^i–l^	4.21 ± 0.02 ^fgh^	15.58 ± 0.09 ^abc^	36.25 ± 0.05 ^f–l^	14.41 ± 0.06 ^p^	131.31	1.96
Tai Htaung Pe Ni	32.02 ± 0.07 ^c^	4.43 ± 0.04 ^d–g^	10.49 ± 0.04 ^i–l^	38.52 ± 0.05 ^bcd^	14.55 ± 0.05 ^op^	131.17	1.74
THA-JSH-2008-81021	29.96 ± 0.04 ^h^	3.44 ± 0.04 ^kl^	11.87 ± 0.05 ^gh^	38.06 ± 0.09 ^b–f^	16.67 ± 0.11 ^c^	138.00	1.99
Tanjingeun	32.13 ± 0.06 ^bc^	3.89 ± 0.02 ^hij^	7.88 ± 0.09 ^m^	38.48 ± 0.17 ^b–e^	17.62 ± 0.16 ^b^	137.71	1.78
Jasaekginggori	32.09 ± 0.11 ^c^	4.27 ± 0.01 ^e–h^	11.66 ± 0.04 ^ghi^	35.19 ± 0.12 ^h–l^	16.79 ± 0.26 ^c^	132.41	1.75
Gatggeungong	32.01 ± 0.01 ^c^	4.16 ± 0.03 ^fgh^	11.26 ± 0.04 ^hij^	36.81 ± 0.12 ^d–j^	15.76 ± 0.11 ^ghi^	132.17	1.76
MMR-JYH-2010-90	31.25 ± 0.07 ^e^	3.99 ± 0.01 ^ghi^	9.76 ± 0.04 ^l^	37.54 ± 0.08 ^b–g^	17.46 ± 0.05 ^b^	137.22	1.84
Sung 99	30.80 ± 0.02 ^f^	3.48 ± 0.00 ^jkl^	11.90 ± 0.01 ^gh^	36.35 ± 0.12 ^f–l^	17.47 ± 0.14 ^b^	136.99	1.92
Guamian hong jiangdou	32.08 ± 0.13 ^c^	4.25 ± 0.01 ^e–h^	10.77 ± 0.06 ^h–l^	37.65 ± 0.03 ^b–g^	15.25 ± 0.05 ^l^	131.81	1.75
Qiu jiang 512	29.93 ± 0.08 ^h^	3.61 ± 0.01 ^ijk^	14.05 ± 0.05 ^de^	34.39 ± 0.06 ^l^	18.02 ± 0.05 ^a^	136.89	1.98
Chungbuk Geosan 2011-18	31.72 ± 0.20 ^d^	4.37 ± 0.02 ^d–g^	11.48 ± 0.12 ^ghi^	35.91 ± 0.09 ^g–l^	16.52 ± 0.15 ^cd^	132.87	1.77
Chungbuk Geosan 2011-245	31.54 ± 0.03 ^d^	4.44 ± 0.02 ^d–g^	11.88 ± 0.12 ^gh^	35.92 ± 0.06 ^g–l^	16.22 ± 0.14 ^ed^	132.38	1.78
901 Zaoshu jiangdou	29.32 ± 0.04l ^mn^	4.79 ± 0.01 ^bcd^	13.84 ± 0.12 ^def^	35.77 ± 0.12 ^g–l^	16.28 ± 0.22 ^de^	134.22	1.93
Techang jinqili	29.25 ± 0.08 ^mn^	4.71 ± 0.01 ^cde^	14.16 ± 0.05 ^de^	35.31 ± 0.10 ^h–l^	16.56 ± 0.09 ^c^	134.46	1.94
Nan 1 Variety	29.64 ± 0.04 ^ijk^	3.34 ± 0.02 ^kl^	13.27 ± 0.07 ^ef^	37.16 ± 0.07 ^c–h^	16.59 ± 0.18 ^c^	137.35	2.03
Hei mei 1	29.14 ± 0.09 ^mn^	4.71 ± 0.01 ^cde^	14.24 ± 0.05 ^de^	36.00 ± 0.06 ^g–l^	15.92 ± 0.19 ^gh^	134.00	1.95
Chunqiu hong jiangdou	32.37 ± 0.04 ^ab^	4.29 ± 0.02 ^e–h^	10.35 ± 0.01 ^i–l^	37.50 ± 0.09 ^b–g^	15.49 ± 0.08 ^jkl^	131.82	1.73
Man di hong wu jia dou	29.75 ± 0.06 ^hi^	3.40 ± 0.02 ^kl^	11.86 ± 0.07 ^gh^	38.37 ± 0.05 ^b–e^	16.63 ± 0.07 ^c^	138.49	2.02
Tichun zhi jiang 28-2	29.08 ± 0.05 ^n^	4.00 ± 0.01 ^ghi^	16.08 ± 0.04 ^ab^	34.82 ± 0.03 ^jkl^	16.01 ± 0.08 ^fg^	133.76	2.02
Te xuan zhang tang wang	29.41 ± 0.02 ^j–m^	4.31 ± 0.01 ^e–h^	16.36 ± 0.07 ^a^	35.05 ± 0.06 ^i–l^	14.88 ± 0.02 ^mn^	131.11	1.97
Hei mei huang zi wang	29.36 ± 0.09 ^lmn^	4.57 ± 0.01 ^c–f^	15.08 ± 0.03 ^bcd^	35.02 ± 0.09 ^jkl^	15.97 ± 0.06 ^fg^	133.04	1.95
Yard long Bean 287/2556	29.14 ± 0.12 ^mn^	4.30 ± 0.03 ^e–h^	12.01 ± 0.18 ^gh^	39.26 ± 0.06 ^b^	15.28 ± 0.12 ^kl^	136.39	1.99
SD 3135	32.04 ± 0.03 ^c^	4.22 ± 0.02 ^fgh^	11.84 ± 0.09 ^gh^	35.91 ± 0.05 ^g–l^	15.99 ± 0.09 ^fg^	131.64	1.76
KSL 170256	31.27 ± 0.05 ^e^	4.60 ± 0.04 ^c–f^	12.62 ± 0.15 ^fg^	35.83 ± 0.08 ^g–l^	15.68 ± 0.09 ^hij^	131.31	1.79
Gyeongnamhabcheon-2019-2	30.33 ± 0.02 ^g^	4.35 ± 0.06 ^d–h^	10.43 ± 0.12 ^i–l^	36.76 ± 0.11 ^d–j^	18.12 ± 0.05 ^a^	138.34	1.88
Range (Min–Max)	29.08–32.45	3.34–5.29	7.88–16.36	34.39–39.26	14.41–18.12	131.11–138.49	1.73–2.03
Total mean	30.60	4.28	12.60	36.58	15.94	-	-
CV (%)	4.18	11.02	16.06	3.36	6.20	-	-

Different superscript letters in a column indicate significantly different mean values between the yardlong bean accessions (*p* < 0.05). DBI: double bond index, TSFA: total saturated fatty acid, TUFA: total unsaturated fatty acid.

**Table 2 antioxidants-13-00463-t002:** Variations in total secondary metabolite contents and antioxidant activities across 35 yardlong bean accessions cultivated in Korea.

Accession Name	Total Metabolite Contents			Antioxidant Activities		
	TPC (mg GAE/g)	TSC (mg DE/g)	TTC (mg CE/g)	DPPH (mg AAE/g)	ABTS (mg TE/g)	RP (mg AAE/g)
San chi lu	4.48 ± 0.07 ^opq^	50.58 ± 2.69 ^efg^	41.69 ± 2.19 ^k^	2.87 ± 0.13 ^j–m^	7.22 ± 0.23 ^mn^	3.19 ± 0.20 ^n–q^
901 Qingjo jiangdou	6.18 ± 0.51 ^e–i^	53.22 ± 1.68 ^d–g^	55.85 ± 5.27 ^jk^	3.66 ± 0.28 ^f–k^	9.73 ± 0.64 ^g–k^	4.82 ± 0.12 ^j–m^
Qing jiang 2006	5.12 ± 0.10 ^jp^	50.22 ± 1.62 ^efg^	46.16 ± 1.83 ^jk^	3.07 ± 0.11 ^h–m^	7.60 ± 0.12 ^k–n^	4.03 ± 0.06 ^l–o^
Gaotian jinguan	4.85 ± 0.22 ^m–p^	35.99 ± 0.66 ^hi^	48.32 ± 2.17 ^jk^	3.30 ± 0.26 ^g–l^	7.37 ± 0.64 ^lmn^	2.68 ± 0.27 ^pq^
Gyeonggiyonginsujib	5.29 ± 0.46 ^i–o^	47.65 ± 2.75 ^fg^	124.29 ± 9.90 ^fgh^	4.36 ± 0.14 ^efg^	10.55 ± 0.76 ^e–i^	7.64 ± 0.56 ^fg^
Gangwonpyeongchang-2003-4	5.75 ± 0.43 ^g–m^	67.26 ± 2.69 ^bc^	141.41 ± 11.48 ^fg^	4.64 ± 0.34 ^def^	11.95 ± 1.18 ^ef^	8.20 ± 1.08 ^def^
Gangwonpyeongchang-2003-17	5.71 ± 0.16 ^g–m^	56.17 ± 3.01 ^def^	107.37 ± 0.39 ^g–j^	5.38 ± 0.70 ^cd^	10.54 ± 0.59 ^e–i^	7.29 ± 1.01 ^fgh^
Hang xin taikong wu jia dou	5.07 ± 0.11 ^k–p^	48.70 ± 4.07 ^efg^	37.38 ± 2.31 ^k^	3.88 ± 0.16 ^f–j^	8.53 ± 0.26 ^i–n^	6.17 ± 0.20 ^hij^
Techang 908 jiangdou	5.56 ± 0.33 ^h–n^	53.22 ± 4.73 ^d–g^	53.99 ± 2.78 ^jk^	3.63 ± 0.09 ^f–k^	8.63 ± 0.83 ^i–n^	6.78 ± 0.32 ^ghi^
Taiwan chunqiu hong	8.15 ± 0.47 ^b^	76.06 ± 3.44 ^ab^	778.34 ± 78.94 ^a^	9.95 ± 0.88 ^a^	18.87 ± 1.21 ^b^	15.58 ± 0.68 ^a^
Techang 902 jiangdou	6.21 ± 0.09 ^e–i^	56.03 ± 6.16 ^def^	47.22 ± 2.41 ^jk^	5.17 ± 0.12 ^cde^	10.89 ± 0.11 ^efg^	7.54 ± 0.10 ^fg^
Tai Htaung Pe Ni	7.13 ± 0.45 ^cde^	75.77 ± 1.93 ^ab^	709.42 ± 57.87 ^b^	7.29 ± 0.63 ^b^	18.06 ± 0.54 ^b^	12.83 ± 0.68 ^b^
THA-JSH-2008-81021	6.32 ± 0.13 ^e–h^	63.51 ± 2.79 ^cd^	590.93 ± 33.66 ^c^	4.16 ± 0.21 ^e–h^	14.06 ± 0.69 ^cd^	7.98 ± 0.27 ^efg^
Tanjingeun	6.39 ± 0.41 ^d–h^	59.18 ± 0.25 ^cde^	50.74 ± 5.40 ^jk^	3.04 ± 0.13 ^i–m^	10.77 ± 0.69 ^e–h^	5.72 ± 0.38 ^ijk^
Jasaekginggori	4.67 ± 0.14 ^n–q^	47.78 ± 1.69 ^fg^	140.20 ± 2.94 ^fg^	3.02 ± 0.06 ^i–m^	9.91 ± 0.36 ^f–j^	4.95 ± 0.21 ^j–m^
Gatggeungong	5.92 ± 0.33 ^g–k^	58.75 ± 4.30 ^cde^	147.23 ± 10.11 ^fg^	3.33 ± 0.19 ^g–l^	11.99 ± 1.38 ^ef^	6.19 ± 0.48 ^hij^
MMR-JYH-2010-90	5.88 ± 0.29 ^g–l^	56.21 ± 2.40 ^def^	71.88 ± 1.96 ^h–k^	2.17 ± 0.18 ^mn^	8.31 ± 0.50 ^j–m^	4.84 ± 0.21 ^j–l^
Sung 99	4.17 ± 0.02 ^pq^	51.01 ± 5.57 ^efg^	31.20 ± 4.61 ^k^	3.14 ± 0.03 ^h–m^	8.73 ± 0.62 ^h–m^	4.57 ± 0.78 ^k–m^
Guamian hong jiangdou	6.64 ± 0.44 ^c–g^	81.01 ± 4.03 ^a^	640.97 ± 52.94 ^c^	6.98 ± 0.48 ^b^	15.38 ± 1.07 ^c^	9.48 ± 0.53 ^cd^
Qiu jiang 512	4.72 ± 0.21 ^n–q^	55.13 ± 5.88 ^def^	39.40 ± 2.64 ^k^	3.12 ± 0.35 ^h–m^	8.49 ± 0.17 ^i–n^	5.04 ± 0.09 ^j–m^
Chungbuk Geosan 2011-18	4.55 ± 0.63 ^opq^	54.22 ± 6.38 ^def^	116.51 ± 15.06 ^ghi^	3.58 ± 0.19 ^f–k^	8.77 ± 1.25 ^h–m^	5.09 ± 0.51 ^j–m^
Chungbuk Geosan 2011-245	3.78 ± 0.02 ^q^	53.83 ± 5.31 ^def^	90.15 ± 13.93 ^g–k^	2.36 ± 0.14l ^mn^	6.67 ± 0.26 ^mn^	4.19 ± 0.11 ^lmn^
901 Zaoshu jiangdou	4.94 ± 0.17 ^l –p^	36.80 ± 1.13 ^hi^	48.50 ± 2.11 ^jk^	3.14 ± 0.39 ^h–m^	6.51 ± 0.15 ^n^	4.19 ± 0.27 ^lmn^
Techang jinqili	6.49 ± 1.03 ^c–h^	49.07 ± 3.90 ^efg^	52.79 ± 5.06 ^jk^	4.15 ± 1.56 ^e–h^	8.61 ± 1.47 ^i–n^	5.40 ± 0.85 ^jkl^
Nan 1 Variety	7.33 ± 0.26 ^bcd^	56.43 ± 8.92 ^def^	529.94 ± 17.07 ^d^	6.00 ± 0.98 ^c^	14.98 ± 1.09 ^c^	9.97 ± 0.68 ^c^
Hei mei 1	7.00 ± 0.14 ^c–f^	53.66 ± 0.40 ^d–g^	60.25 ± 1.37 ^ijk^	4.14 ± 0.11 ^e–h^	9.43 ± 0.35 ^g–l^	6.02 ± 0.06 ^hij^
Chunqiu hong jiangdou	6.59 ± 0.80 ^c–g^	57.84 ± 7.49 ^c–f^	619.60 ± 92.35 ^c^	6.97 ± 0.72 ^b^	12.52 ± 1.65 ^de^	9.13 ± 1.54 ^cde^
Man di hong wu jia dou	9.13 ± 0.21 ^a^	82.55 ± 10.54 ^a^	334.63 ± 10.23 ^e^	9.27 ± 1.43 ^a^	21.21 ± 1.90 ^a^	14.69 ± 1.11 ^a^
Tichun zhi jiang28-2	5.80 ± 0.26 ^g–m^	49.74 ± 3.15 ^efg^	33.41 ± 3.01 ^k^	4.10 ± 0.34 ^f–i^	8.43 ± 0.66 ^i–m^	5.31 ± 0.37 ^jkl^
Te xuan zhang tang wang	4.94 ± 0.49 ^l–p^	27.42 ± 3.86 ^ij^	31.98 ± 2.14 ^k^	1.66 ± 0.14 ^n^	6.78 ± 0.62 ^mn^	2.02 ± 0.24 ^q^
Hei mei huang zi wang	6.52 ± 0.88 ^c–h^	43.07 ± 5.77 ^gh^	52.23 ± 6.66 ^jk^	3.18 ± 0.52 ^h–m^	12.56 ± 1.94 ^de^	4.21 ± 0.37 ^lmn^
Yard long Bean 287/2556	7.41 ± 0.41 ^bc^	56.95 ± 4.78 ^def^	326.49 ± 25.13 ^e^	5.62 ± 0.39 ^cd^	18.30 ± 0.80 ^b^	9.45 ± 1.19 ^cd^
SD 3135	6.05 ± 0.28 ^f–j^	47.17 ± 1.41 ^fg^	179.53 ± 10.31 ^f^	3.95 ± 0.18 ^f–j^	12.32 ± 0.86 ^de^	6.07 ± 0.58 ^hij^
KSL 170256	4.58 ± 0.15 ^opq^	30.20 ± 2.33 ^ij^	141.37 ± 15.83 ^fg^	2.67 ± 0.27 ^klm^	7.79 ± 0.70 ^j–n^	3.67 ± 0.35 ^m–p^
Gyeongnamhabcheon-2019-2	4.47 ± 0.63 ^opq^	25.79 ± 4.69 ^j^	43.54 ± 5.08 ^k^	1.63 ± 0.13 ^n^	6.70 ± 0.92 ^mn^	2.76 ± 0.26 ^opq^
Range (Min- Max)	3.78–9.13	25.79–82.55	31.20–778.34	1.63–9.95	6.51–21.21	2.02–15.58
Total mean	5.82	53.38	187.57	4.25	10.83	6.51
CV (%)	20.01	24.31	118.36	44.92	34.86	48.17

Different superscript letters in a column indicate significantly different mean values between the yardlong bean accessions (*p* < 0.05). ABTS: ABTS^•+^ scavenging activity, DPPH: DPPH^•^ scavenging activity, RP: reducing power, TPC: total phenolic content, TSC: total saponin content, TTC: total tannin content.

**Table 3 antioxidants-13-00463-t003:** Variations in agronomic traits, biochemical contents, and antioxidant activities according to genotype and origin.

Category	Variable	Genotype		Origin			
		Cultivar	Landrace	Thailand	Myanmar	China	Korea
Agronomic traits	DF (days)	53.50 ± 3.04 ^b^	55.87 ± 2.73 ^a^	56.33 ± 2.05 ^a^	56.67 ± 2.05 ^a^	52.56 ± 2.89 ^b^	56.64 ± 1.55 ^a^
	DFM (days)	21.60 ± 3.67 ^a^	20.47 ± 3.77 ^a^	25.67 ± 2.49 ^a^	23.33 ± 5.91 ^ab^	21.22 ± 3.26 ^ab^	19.09 ± 2.27 ^b^
	DM (days)	75.10 ± 4.68 ^a^	76.33 ± 4.83 ^a^	82.00 ± 4.24 ^a^	80.00 ± 7.87 ^ab^	73.78 ± 3.49 ^c^	75.73 ± 3.02 ^bc^
	PL (cm)	51.96 ± 11.63 ^a^	47.79 ± 9.96 ^a^	44.47 ± 11.45 ^ab^	50.53 ± 9.54 ^ab^	55.36 ± 11.38 ^a^	43.15 ± 4.48 ^b^
	SPP (n)	16.04 ± 1.98 ^a^	15.43 ± 1.86 ^a^	14.67 ± 2.50 ^a^	13.73 ± 1.54 ^a^	16.14 ± 1.59 ^a^	16.04 ± 1.97 ^a^
	HSW (g)	15.97 ± 2.60 ^a^	15.71 ± 2.63 ^a^	17.79 ± 3.93 a	15.70 ± 2.48 ^a^	15.59 ± 2.11 ^a^	15.82 ± 2.73 ^a^
Nutritional components	TP (%)	25.65 ± 1.20 ^a^	25.92 ± 0.89 ^a^	26.57 ± 1.11 ^a^	26.36 ± 0.94 ^a^	25.45 ± 1.09 ^a^	25.90 ± 0.87 ^a^
	TF (%)	1.33 ± 0.34 ^a^	1.09 ± 0.30 ^b^	1.08 ± 0.40 ^b^	1.01 ± 0.07 ^b^	1.44 ± 0.31 ^a^	0.97 ± 0.14 ^b^
	CFC (%)	4.96 ± 0.51 ^a^	4.87 ± 0.59 ^a^	4.92 ± 0.67 ^a^	5.03 ± 0.04 ^a^	5.05 ± 0.42 ^a^	4.68 ± 0.66 ^a^
	DFC (%)	19.32 ± 1.33 ^a^	19.27 ± 1.57 ^a^	19.71 ± 0.65 ^a^	19.26 ± 1.12 ^a^	19.27 ± 1.42 ^a^	19.24 ± 1.67 ^a^
	Vit C (mg/g)	2.08 ± 0.60 ^a^	2.11 ± 0.65 ^a^	1.94 ± 0.10 ^a^	1.93 ± 0.27 ^a^	2.01 ± 0.76 ^a^	2.32 ± 0.44 ^a^
Fatty acids	PA (%)	30.02 ± 1.16 ^b^	31.39 ± 0.98 ^a^	29.58 ± 0.34 ^b^	31.36 ± 0.50 ^a^	29.89 ± 1.10 ^b^	31.84 ± 0.59 ^a^
	SA (%)	4.31 ± 0.56 ^a^	4.25 ± 0.32 ^a^	3.69 ± 0.43 ^b^	3.97 ± 0.39 ^ab^	4.47 ± 0.49 ^a^	4.21 ± 0.21 ^ab^
	OA (%)	13.42 ± 1.93 ^a^	11.50 ± 1.58 ^b^	12.39 ± 0.63 ^ab^	10.72 ± 0.89 ^b^	13.84 ± 0.1.86 ^a^	11.13 ± 1.16 ^b^
	LA (%)	36.60 ± 1.40 ^a^	36.55 ± 0.95 ^a^	38.16 ± 0.86 ^a^	37.47 ± 0.89 ^ab^	36.23 ± 1.25 ^b^	36.48 ± 0.86 ^b^
	LLA (%)	15.65 ± 0.91 ^b^	16.32 ± 0.96 ^a^	16.18 ± 0.64 ^a^	16.49 ± 1.37 ^a^	15.56 ± 0.89 ^a^	16.34 ± 0.82 ^a^
	TSFA (%)	34.33 ± 1.17 ^b^	35.63 ± 0.88 ^a^	33.27 ± 0.21 ^c^	35.32 ± 0.89 ^ab^	34.37 ± 1.06 ^bc^	36.06 ± 0.47 ^a^
	TUFA (%)	65.67 ± 1.17 ^a^	64.37 ± 0.88 ^b^	66.73 ± 0.21 ^a^	64.68 ± 0.89 ^bc^	65.63 ± 1.06 ^ab^	63.95 ± 0.47 ^c^
	PUFA (%)	52.25 ± 1.46 ^a^	52.87 ± 1.39 ^a^	54.34 ± 0.43 ^a^	53.96 ± 0.80 ^a^	51.79 ± 1.21 ^b^	52.82 ± 1.33 ^a^
Total secondary metabolites	TPC (mg GAE/g)	6.18 ± 1.14 ^a^	5.34 ± 1.01 ^b^	7.02 ± 0.50 ^a^	5.73 ± 1.21 ^ab^	6.02 ± 1.21 ^ab^	5.20 ± 0.79 ^b^
	TTC (mg CE/g)	223.78 ± 253.46 ^a^	139.30 ± 158.94 ^a^	482.45 ± 113.06 ^a^	270.84 ± 310.57 ^ab^	167.93 ± 39.55 ^b^	116.58 ± 3.22 ^b^
	TSC (mg DE/g)	54.46 ± 13.58 ^a^	51.94 ± 11.97 ^a^	58.96 ± 3.22 ^a^	61.00 ± 10.66 ^a^	53.35 ± 14.12 ^a^	49.82 ± 11.78 ^a^
Antioxidant activities	DPPH (mg AAE/g)	4.82 ± 2.08 ^a^	3.48 ± 1.30 ^b^	5.26 ± 0.79 ^a^	4.20 ± 2.22 ^a^	4.57 ± 2.20 ^a^	3.45 ± 1.03 ^a^
	RP (mg AAE/g)	7.09 ± 3.48 ^a^	5.73 ± 2.39 ^a^	9.13 ± 1.82 ^a^	7.41 ± 4.50 ^a^	6.46 ± 4.06 ^a^	5.62 ± 1.96 ^a^
	ABTS (mg TE/g)	11.504.23 ^a^	9.94 ± 2.83 ^a^	15.78 ± 0.84 ^a^	11.70 ± 3.83 ^ab^	10.49 ± 3.63 ^b^	9.81 ± 1.62 ^b^

Different superscript letters across a row within a category represent significantly different mean values (*p* < 0.05). ABTS: ABTS^•+^ scavenging activity, CFC: crude fiber content, DF: days to flowering, DFC: dietary fiber content, DFM: days from flowering to maturity, DM: days to maturity, DPPH: DPPH^•^ scavenging activity, HSW: one-hundred seeds weight, LA: linoleic acid, LLA: linolenic acid, OA: oleic acid, PA: palmitic acid, PL: pod length, PUAF: total polyunsaturated fatty acid, RP: reducing power, SA: stearic acid, SPP; number of seeds per pod, TF: total fat content, TP: total protein content, TSFA: total saturated fatty acid, TUFA: total unsaturated fatty acid, TPC: total phenolic content, TSC: total saponin content, TTC: total tannin content, Vit C: vitamin C content.

## Data Availability

All the data related to this study are incorporated in the manuscript and Appendix A. Further inquiries can be directed to the first author.

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
