# Peer review of "Multivariate Analysis of Biochemical Properties Reveals Diversity among Yardlong Beans of Different Origins"

_antioxidants, 2024, doi:10.3390/antiox13040463_

Round 1

Reviewer 1 Report

The paper entitled „Multivariate Analysis of Biochemical Properties Reveals Diversity among Yardlong beans of Different Origins” aimed to analysis of the nutrient levels, secondary metabolite contents, and antioxidant activities of 35 yardlong bean accessions from China, Korea, Myanmar, and Thailand, along with their key agronomic traits. The issue is interesting and important. In my opinion, paper is well prepared. Authors decided to perform various studies along with their detail statistical analysis. The results were presented clearly. The paper can be accepted to publication after the minor revision. Please see below:

Abstract: informative and readable

Introduction: informative. Authors characterized the Vigna unguiculata along with its pro-health properties. The information are based on up to date references.

Materials and methods:

·         - in my opinion, Authors should provide full information (2.1. Chemicals and reagents ) about all reagents used in the experiments

·         - Were conditions provided to prevent accidental cross-pollination of plants, which led to the mixing/modification of the tested species?

Results: obtained results are interesting and significant. Scientists allowed to compare yardlong bean from China, Korea, Myanmar, and Thailand in terms of nutrient levels, secondary metabolite contents, and antioxidant activities. The results are well described and presented in form of numerous figures. Additionally Authors presents Pearson’s correlation matrix showing the association between the analyzed quantitative parameters.

Discussion: whereas Results are analyzed in detail, Discussion is not satisfacotry. In my opinion, Authors should provide more information about similar studies performed in past by other scientists. Please provide separate subsection for it.

Conclusions: sufficient

References: 60 possitions, adequate and up-to-date

All Figures and Tables are clearly presented. There is no specific comments to this part of paper. PLease consider Major comments.

Reviewer 2 Report

The manuscript deals with the analysis of several varieties of Vigna sp, a widely used and consumed legume.

35 varieties of Vigna sp (Yardlong bean or asparagus bean) from China, Korea, Myanmar and Thailand were analysed for their different nutritional components such as: crude fiber, dietary fiber, total fat, total protein, vitamin C contents and fatty acids.  Within the secondary metabolites, the content of total phenols, total saponins and total tannins were assessed. The authors also evaluated the in vitro antioxidant activities by means of DPPH scavenging activity, reducing power and by the ABTS method. In addition, differences between areas of origin and genotype were determined. An extensive multivariate statistical analysis with hierarchical cluster analysis (HCA) and principal component analysis (PCA) was applied.

The manuscript is sound and the arguments appear well supported by experiments and references. The results are well discussed and the conclusions are in line with the work plan.

Among the anti-nutritional factors, only tannins were determined. To complete the study, the authors can include why they did not also determine L-DOPA, hydrogen cyanide and phytic acid as reported in (DOI: 10.1111/j.1745-4514.2005.00014.x). An explanation is enough.

The manuscript can be accepted after minor revisions.

The Figures and Tables are clear and comprehensive. They are adequate in number.

Statistical analysis has been carried out and the results are clearly discussed.

The supplementary material supports the discussion

References support the arguments presented and are adequate and relevant in the discussion.

Reviewer 3 Report

The manuscript analyzed the crude fiber, dietary fiber, total protein, total fat contents, and antioxidant activities of Yardlong beans from different origins. Several points need to be addressed:

Lines 96, 174, 240...: in order to keep the research paper objective, subjective words like  "we", and "our" should be avoided and it is recommended to use the passive tense, please revise the whole manuscript. 

Line 178: please use the equation form.

Line 194: How the authors measure the pod colour and seed colour are missing. 

Lines 236-239: Those did not contribute to the discussion too much. It will be better to explain the reason why there are different CFC and DFC for Yardlong beans from different places. The mechanism will attract the reader more than describing a matter of fact. For the following sections, it should be the same, explain the reason for the observation. 

Figure 1: I suggest the authors separating the total fat and total protein content because they are different categories and it is very hard to find differences between different accessions concerning total fat.  Line 249 "B" should be "b". For (a) and (c), I suggest putting the specific caption on the x-axis, rather than only "content"

Line 289: ΣTUFA:ΣTSFA have not been defined in the main text, if TSFA is "total saturated fatty acid", what the differences between ΣTSFA and TSFA? 

Figure 6 for TSFA in the third column, it is hard to read the characteries, please revise accordingly.

The manuscript analyzed the crude fiber, dietary fiber, total protein, total fat contents, and antioxidant activities of Yardlong beans from different origins. Several points need to be addressed:

Lines 96, 174, 240...: in order to keep the research paper objective, subjective words like  "we", and "our" should be avoided and it is recommended to use the passive tense, please revise the whole manuscript. 

Line 178: please use the equation form.

Line 194: How the authors measure the pod colour and seed colour are missing. 

Lines 236-239: Those did not contribute to the discussion too much. It will be better to explain the reason why there are different CFC and DFC for Yardlong beans from different places. The mechanism will attract the reader more than describing a matter of fact. For the following sections, it should be the same, explain the reason for the observation. 

Figure 1: I suggest the authors separating the total fat and total protein content because they are different categories and it is very hard to find differences between different accessions concerning total fat.  Line 249 "B" should be "b". For (a) and (c), I suggest putting the specific caption on the x-axis, rather than only "content"

Line 289: ΣTUFA:ΣTSFA have not been defined in the main text, if TSFA is "total saturated fatty acid", what the differences between ΣTSFA and TSFA? 

Figure 6 for TSFA in the third column, it is hard to read the characteries, please revise accordingly.

Reviewer 4 Report

This is a research article about biochemical properties of different origin Yardlong beans. Detailed multivariate analysis for performed for this research. This manuscript, in its structure and content, fits with the subject matter of this journal. The content of the manuscript is informative in both scientific and practical terms.
The manuscript is based on 60 references from year 1987 to 2023. A number of literature sources (~50%, 29/60) are older than 5 years, so I suggest updating the bibliography with newer sources (as much as it is possible for the authors) with the latest scientific discoveries in the field.
The figures in the manuscript illustrate the topic well and fit well into the context.
Table 3 lacks standart deviation or standard error values, which, in my opinion, would provide valuable additional information for the dissemination of the study results.
The research design and methodology do not raise any questions for me, so I have no further comments.

I don't have any additional detailed comments, everything is described in major comments section.

Reviewer 5 Report

-        -   In the paragraph „The fatty acids were identified by comparing retention times to their corresponding external standards.......”  you specified the use of standards. Which standards are used?

-          Antioxidant activity was assessed by three assays: DPPH, ABTS si RP.  How do you explain the large differences between the values obtained using the three tests for the same product?

-       

-          Two chromatographic techniques were used in the paper: GC-FID for fatty acids and HPLC for vitamin C. Examples of chromatograms for the identified compounds should be presented (as additional material).

-      

Round 2

Reviewer 2 Report

Accept in the present form

Accept in the present form

Author Response

Dear Reviewer,

Thank you for the insightful feedback you shared during the review process. It is great to hear that all of your suggestions and comments have been fully addressed.

Reviewer 3 Report

The authors addressed some of my comments, but there is still some need to improve:

Line 187: equations should be marked with a number, please check throughout the manuscript and use the template from the journal.

Figure 6 is not improved. As can be found in the third column and 5th row of TSFA, the dark blue colour overlaps the number and is hard to read. 

Moreover, please reply to the comments according to line number, it wastes the reviewers' time to "guess" where your answer could be: for example, "To avoid redundancy of ideas, a general description regarding the possible factors that cause such variations is now described in section 3.7." it will be more helpful if you can mention with line number. Please be careful for future submissions. 

The authors addressed some of my comments, but there is still some need to improve:

Line 187: equations should be marked with a number, please check throughout the manuscript and use the template from the journal.

Figure 6 is not improved. As can be found in the third column and 5th row of TSFA, the dark blue colour overlaps the number and is hard to read. 

Moreover, please reply to the comments according to line number, it wastes the reviewers' time to "guess" where your answer could be: for example, "To avoid redundancy of ideas, a general description regarding the possible factors that cause such variations is now described in section 3.7." it will be more helpful if you can mention with line number. Please be careful for future submissions. 

Reviewer 5 Report

The authors responded to the observations.

The authors have added the necessary figures (supplement doc).

Author Response

Dear Reviewer,

Thank you for the insightful and professional feedback you shared during the review process. It is great to hear that all of your suggestions and comments have been fully addressed.

Round 3

Reviewer 3 Report

The authors did not address my previous comment to explain the possible factors that cause variations in section 3.7. Where is your answer? It is neither marked in the main text nor answer with the line number!!!

The authors did not address my previous comment to explain the possible factors that cause variations in section 3.7. Where is your answer? It is neither marked in the main text nor answer with the line number!!!

Author Response

We apologize for the mis-communication that occurred. In Reviewer 3 last comment, we thought that the reviewer (Reviewer 3) advised us to consider such cases (describing responses in accordance with line numbers rather than sections) during future submissions.   Now we have addressed the issue, and please find attached the point-by-point responses to Reviewer 3 comments. Specifically, line numbers regarding the third comment (about the factors affecting the biochemical factors) are included. Attached is also the latest manuscript both in pdf (clear version) and word (track-change) format. 

Round 4

Reviewer 3 Report

The authors have addressed my comments.

The authors have addressed my comments.